# Molecular surveillance of hepatitis E virus in wastewater in Yaoundé, Cameroon

Dowbiss Meta-Djomsi[1,2☯]*, Marie Atsama-Amougou[1,3☯], Modeste Romuald Ngamaleu[1,4,5], Celestin Godwe[1‡], Martin Maidadi-Foudi[1‡], Marcel Tongo[1], Joseph Fokam[5,6,7], Charles Kouanfack[1,8,9], Ahidjo Ayouba[1,10]

1 Centre de Recherche sur les Maladies Émergentes et Re-Emergentes, Yaounde, Cameroon, 2 Department of microbiology, Faculty of Sciences, University of Buea, Buea, Cameroon, 3 Laboratory of Pharmacology and Toxicology, University of Yaounde I, Yaounde, Cameroon, 4 Biotechnology Center, University of Yaounde I, Yaounde, Cameroon, 5 School of Health Sciences, Catholic University of Central Africa, Yaounde, Cameroon, 6 Central Technical Group, National AIDS Control Committee (NACC), Yaoundé, Cameroon, 7 Virology Laboratory, Chantal BIYA International Reference Centre for Research on HIV/AIDS Prevention and Management, Yaoundé, Cameroon, 8 Faculty of Medicine and Pharmaceutical Sciences, University of Dschang, Dschang, Cameroon, 9 Day Care Hospital, Central Hospital of Yaounde, Yaounde, Cameroon, 10 TransVIHMI, University of Montpellier, Inserm, Institut de Recherche pour le Developpement, Montpellier, France

☯ These authors contributed equally to this work.
‡ These authors also contributed equally to this work.
* medjodow@yahoo.fr

## Abstract

### Background

Hepatitis E virus (HEV) is a zoonotic pathogen mainly transmitted through contaminated food or water in sub-Saharan countries, highlighting the need for environmental surveillance. This study aimed to assess the burden and molecular characterization of HEV in environmental wastewater.

### Method

A community-based surveillance was conducted in Yaoundé, Cameroon, using untreated wastewater samples collected monthly from January to December 2023 from hospitals, residential sewage systems, markets, and plant watering points. Molecular phylogeny was performed on sequences from the Open Reading Frame 1 region.

### Results

HEV was detected in 26.4% (19/72) of all sites, with prevalence ranging from 8.3% (1/12) in hospitals to 41.7% (5/12) in residential areas (p = 0.0022). Of the 19 positives, HEV detection was highest in specific residential areas (26.3%) and plant watering points (15.8%). Detection rates were significantly higher during the short dry season (36.8%) and long dry season (31.6%) compared to the short-wet season (21.05%) and long-wet season (10.53%) (p = 0.034). Phylogenetic analysis of the

**Data availability statement:** All relevant data are within the manuscript and its Supporting Information files.

**Funding:** The author(s) received no specific funding for this work.

**Competing interests:** The authors have declared that no competing interests exist.

sequenced samples revealed that the detected HEV strains are closely related to Orthohepevirus C genotype C1 previously associated with rodents rather than to the classical human HEV genotypes. This finding raises important questions about possible zoonotic transmission in densely populated urban areas.

## Conclusion

This is the first study to report HEV detection and genetic analysis in wastewater from the Mfoundi Division of Yaoundé, and only the second such report in Cameroon. The presence of HEV in community wastewater, especially from residential and irrigation sites, suggests widespread circulation and potential environmental and foodborne risks. The identification of HEV-C1-like strains highlights the possible role of rodents in transmission. These findings emphasize the importance of integrating wastewater surveillance into public health strategies and call for further research on zoonotic sources through a One Health lens.

## Introduction

The enteric zoonotic pathogen known as the hepatitis E virus (HEV) can cause either acute or chronic viral hepatitis in humans. It is primally transmitted through contaminated water and contaminated food, or by direct contact with infected animals [1–4]. Poor sanitation in developing countries frequently contaminates water supplies with waste [5–7]. The World Health Organization (WHO) estimates that there are approximately 20 million cases of HEV annually, leading to more than 44,000 deaths and 3,000 stillbirths globally [8].

The hepatitis E virus (HEV) is a non-enveloped, single-stranded, positive-sense RNA virus belonging to the Hepeviridae family, and the Paslahepevirus genus. Its genome is approximately 7.2 kilobases (kb) in length and comprises three open reading frames (ORFs). ORF1 encodes non-structural proteins involved in viral replication, ORF2 encodes the viral capsid protein, and ORF3 encodes a small multifunctional protein associated with virus egress and host interaction [9–11]. Among the eight identified HEV genotypes (HEV-1 to HEV-8), five (HEV-1, HEV-2, HEV-3, HEV-4 and HEV-7) are known to infect humans. HEV genotype 3 (HEV-3) is zoonotic and widely distributed among both humans and animals, particularly in developed and transitional countries. The primary animal hosts of HEV-3 include pigs, rabbits, deer, and mongooses. HEV-3 is further subdivided into multiple subtypes (e.g., 3a to 3j) based on phylogenetic differences. Similarly, HEV genotype 4 (HEV-4) infects both humans and pigs. Genotypes 3, 4, and 7 exhibit zoonotic potential. Genotypes 5 and 6 have been identified in wild boars, while genotypes 7 and 8 have recently been isolated from camelids and dromedaries [9,11].

The distribution of HEV varies globally. HEV-1 is mainly found in South and Central Asia and North Africa, while HEV-2 is primarily found in Central America and West Africa. HEV-3 is widespread, and HEV-4 is mainly found in Asia and Europe. HEV-1 and HEV-2 are associated with high mortality rates, particularly during the third

trimester of pregnancy, and can increase the risk of acute liver failure. In contrast, HEV-3 and HEV-4 are more prevalent in industrialized countries and can cause persistent infections in immunocompromised individuals, which may subsequently lead to accelerated progression toward liver cirrhosis. The primary modes of virus transmission are through contact with infected animals or through consumption of infected animal products or contaminated food and drink [12–16].

HEV spreads through the faecal-oral route. In less developed countries, it is commonly considered a waterborne disease, as it is often transmitted through contaminated water due to poor sanitation. However, in more developed countries, it can also be contracted by eating undercooked, contaminated meat, particularly pork [3,14,17,18]. The presence of HEV in drinking and irrigation water is a growing public health concern, as it can contaminate these vital resources. HEV has been detected in wastewater samples from several countries [16]. Previous studies have identified genotypes 1, 3, and 4 in humans and pigs, establishing Cameroon as an endemic country for HEV [19–21]. However, there is limited information on the prevalence and diversity of HEV in Cameroonian wastewater. Currently, only preliminary data is available on HEV in environmental sewage is available from the northern region, where HEV genotype 3, subtype 3a, has been detected [19]. Our aim is to evaluate the prevalence and molecular characteristics of circulating HEV strains in environmental wastewater in Yaoundé, Cameroon, by analyzing wastewater samples collected from various sites, including hospitals, markets, residential areas and irrigation systems.

## Materials and methods

### Study sites

This research was carried out in Yaoundé, the capital city of Cameroon, located in the Mfoundi department of the central region as described in our recent article [22]. The city covers an area of about 256 km² and is situated between latitudes 4°45'N and 4°00'N, and longitudes 11°20'E and 11°40'E. Yaoundé has a tropical climate with two main seasons: dry and wet. The city experiences a long dry season from December to February, followed by a shorter one in July and August. The rainy periods are split into two: from March to June and again from September to November. This pattern of alternating dry and wet seasons is typical of the region.

### Inclusivity in global research

As a national reference laboratory for emerging and re-emerging epidemic preparedness and response, our work focuses on supporting public health efforts (S1 file). No permits were required to access the field sites.

### Sampling and concentration

Untreated wastewater samples were collected monthly from various locations, including the hospital (CHUY_university hospital center), residential sewage systems (BONAS, PAPOSY, Biyem Assi, Mvog Ada), and factory water points (Diderot) [22]. The samples were taken early in the morning, between 6:00 and 6:30, using sterile tubes that were kept at temperatures between 4°C and 8°C and transported to the laboratory within six hours. In the laboratory, the samples were concentrated using the polyethylene glycol (PEG) precipitation method as described by Greening et al. in 2022 and modified by Fatawou et al. in 2023 [19,23]. Briefly, 500 mL of wastewater was pre-treated by centrifugation at 1,940 g and 4 °C for 30 minutes to minimise the presence of PCR inhibitors. The resulting supernatant was then chemically treated using a 1 mol/L NaOH solution. Next, a mixture of 39.5 mL of 22% dextran, 287 mL of 29% PEG 6000, and 35 mL of 5 M NaCl solution was added. The resulting pellet was treated with chloroform to obtain a clear sample, and Gentamicin 100 mg/mL reduce bacterial contamination.

### Molecular detection and characterization of HEV

Total DNA/RNA was extracted from 200 µL of concentrated wastewater using a commercial purification kit with an inhibitor removal spin column (DaAnGene®), following the manufacturer's instructions. 50 µL of elution buffer was used to

elute the isolated nucleic acids, which were then kept at −80°C for additional analysis. The partial HEV ORF1 (Open Reading Frame 1) gene fragment (440 bp) was amplified using a heminested PCR-based assay, as previously described [4]. The extracted nucleic acid was reverse transcribed into cDNA using 10 μL of RNA and random hexamers, with the RevertAid First Strand cDNA Synthesis kit (Thermo Fisher Scientific) according to the manufacturer's protocol. First and second-round PCR was performed using the HotStarTaq PCR kit (QIAGEN, Germany) and denatured primers DE-F4228 (ACYTTYTGTGCYYTITTT GGTCCITGGTT), DE-R4598 (CCGGGTTCRCCIGAGTGTTTCTTCCA), and DE-R4565 (GCCATGTTCCAGAYGGTGTTCCA) [24]. Gel electrophoresis on 1% agarose gels was used to separate the PCR products. Amplicons of approximately 440 bp from the second round of PCR were considered positive.

## HEV Genotyping

HEV genotyping was performed by comparing Cameroonian samples with strains from other geographic regions available in GenBank, using sequencing and phylogenetic analysis.

Phylogenetic trees were constructed using the neighbor-joining method and the Kimura two-parameter model in the MEGA 10 software, based on the nucleotide sequences of the amplified gene. Sequencing was conducted using the 3730XL DNA Analyzer (Applied Biosystems). To ensure the accuracy of the phylogenetic tree analysis, bootstrap resampling was performed 1,000 times. The sequences obtained for this study were uploaded to GenBank with accession numbers PP764563 to PP764569.

## Statistical analysis

Statistical analyses were performed using IBM SPSS Statistics 25.0 software. Qualitative variables were presented as frequencies, and the chi-square test was used to compare proportions, with a significance level set at p values <0.05".

## Results

### HEV detection

A total of 72 wastewater samples were collected from January to December 2023. These included 12 samples from the CHUY_ university hospital center, 12 from the Diderot factory, 12 from the Mvog Ada market, and 36 from three residential wastewater treatment systems in the Mfoundi department (Bonas, Paposy, and Biyem Assi), with 12 samples taken from each site.

As shown in Table 1, HEV was detected in 26.4% (19 out of 72) of wastewater samples. Notably, HEV was identified in samples from all six study locations. The highest detection rates were observed in Bonas and Paposy, at 41.7% (05 out of 12 samples each), while the lowest rate was recorded in CHUY at 8.4% (one out of 12 samples). Biyem Assi and Diderot showed detection rates of 25% (03 out of 12 samples), while Mvog Ada's rate was 16.7% (two out of 12 samples).

**Table 1. Summary of hepatitis E virus detection at each collection site.**

| Sites | N tested | N detected | % of detection |
|---|---|---|---|
| Biyem Assi | 12 | 3 | 25 |
| Bonas | 12 | 5 | 41.7 |
| CHUY | 12 | 1 | 8.4 |
| Diderot | 12 | 3 | 25 |
| Mvogada | 12 | 2 | 16.7 |
| Paposy | 12 | 5 | 41.7 |
| Total | 72 | 19 | 26.4 |

*N=Total number of samples, %= percentage.*

Detection rates varied by source type. HEV was detected significantly more frequently in wastewater from residential areas than in wastewater from other sites (p = 0.0022), followed by wastewater from markets and then from hospitals. Specifically, HEV was found in wastewater from the CHUY hospital at a rate of 8.4% (1/12), from the Mvog Ada market at 16.7% (2/12) and from the Diderot plant watering point at 25% (3/12). In contrast, residential wastewater showed higher positivity rates: 25% (3/12) in Biyem Assi and 41.7% (5/12) in both Paposy and Bonas.

In this study, 52.6% (10/19) of the HEV-detectable samples exhibited a robust signal under UV illumination after electrophoresis, suggesting a higher concentration of HEV RNA. The signal intensity was assessed visually on the gel and scored on a scale from 1 to 3, where 1 indicates the weakest signal and 3 indicates the strongest. Of the samples that tested strongly positive, 30% (3/10) were from BONAS, 10% (1/10) from CHUY, 20% (2/10) from DIDEROT, 30% (3/10) from PAPOSY and 10% (1/10) from BIYEMASSI. The HEV signal detections were found at the plant watering point and in effluent from domestic sewage systems.

## Seasonal distribution of HEV

The selected sites were monitored for a full year (January to December) to determine whether seasonal factors influenced the significant variation in HEV detection in wastewater. This study observed seasonal differences in HEV distribution. The highest HEV detection rates overall were recorded during the short and long dry seasons, at 36.8% (7/19) and 31.6% (6/19), respectively. By contrast, the long-wet season had a detection rate of 21.05% (4/19), while the extended wet season had the lowest rate at 10.53% (2/19) (p = 0.034). Notably, effluent from BONAS consistently showed the highest HEV detection across all seasons (Fig. 1).

## Phylogenetic analysis

The results of the phylogenetic analysis are shown in Fig 2. Out of the 10 positive samples with strong HEV RNA signals detected by nested RT-PCR, only seven (70%) produced good sequencing results. The nucleotide sequences of these seven HEV strains have been uploaded to GenBank and can be accessed using the following accession numbers: PP764563 to PP764569 (S2 table). This study involved conducting a phylogenetic analysis of HEV strains identified in various types of wastewater in the Centre Region of Cameroon, specifically in Yaoundé. This analysis compared a highly conserved 440 bp region of ORF1 with HEV strains from different geographical areas that were available in GenBank. Phylogenetic analysis of the seven sequences indicated that they all clustered with HEV-3.

## Discussion

This is the second report of the detection of the hepatitis E virus (HEV) genome in environmental wastewater samples in Cameroon. HEV was first detected and characterized in sewage environmental samples of the North Region in 2023 [19]. This study presents the first report on the detection and characterization of HEV in wastewater from the Mfoundi Division in Cameroon's Centre Region. The present study provides information on the circulation of HEV in the untreated wastewater of a hospital (CHUY), a residential area (Bonas, Paposy, Biyem Assi), a market (Mvog Ada) and a plant watering point (Diderot).

In the current work, HEV was detected in all six of the selected sites with an overall detection rate of 26.4% and the detection rates of 5.3%, 10.5%, 15.8% and 26.3% for CHUY, Mvog Ada, Biyem Assi and Paposy, respectively. Consistent with other studies, we observed a higher prevalence of HEV in various types of untreated wastewater in the Mfoundi Division. Our results suggest widespread circulation of HEV in Yaoundé communities in Cameroon. Our findings are consistent with data reported in many other studies [15,16,25–27]. Notably, a high HEV detection rate was observed in wastewater from residential systems, as we found, 68% (13/19) of the samples tested positive. This observation highlights that the presence of HEV in the environment resulting from viral excretion by infected individuals may contribute to exposure risk in Cameroon. It suggests that untreated wastewater can serve as a useful tool for monitoring HEV and other enteric pathogens within the community, even during non-epidemic periods.

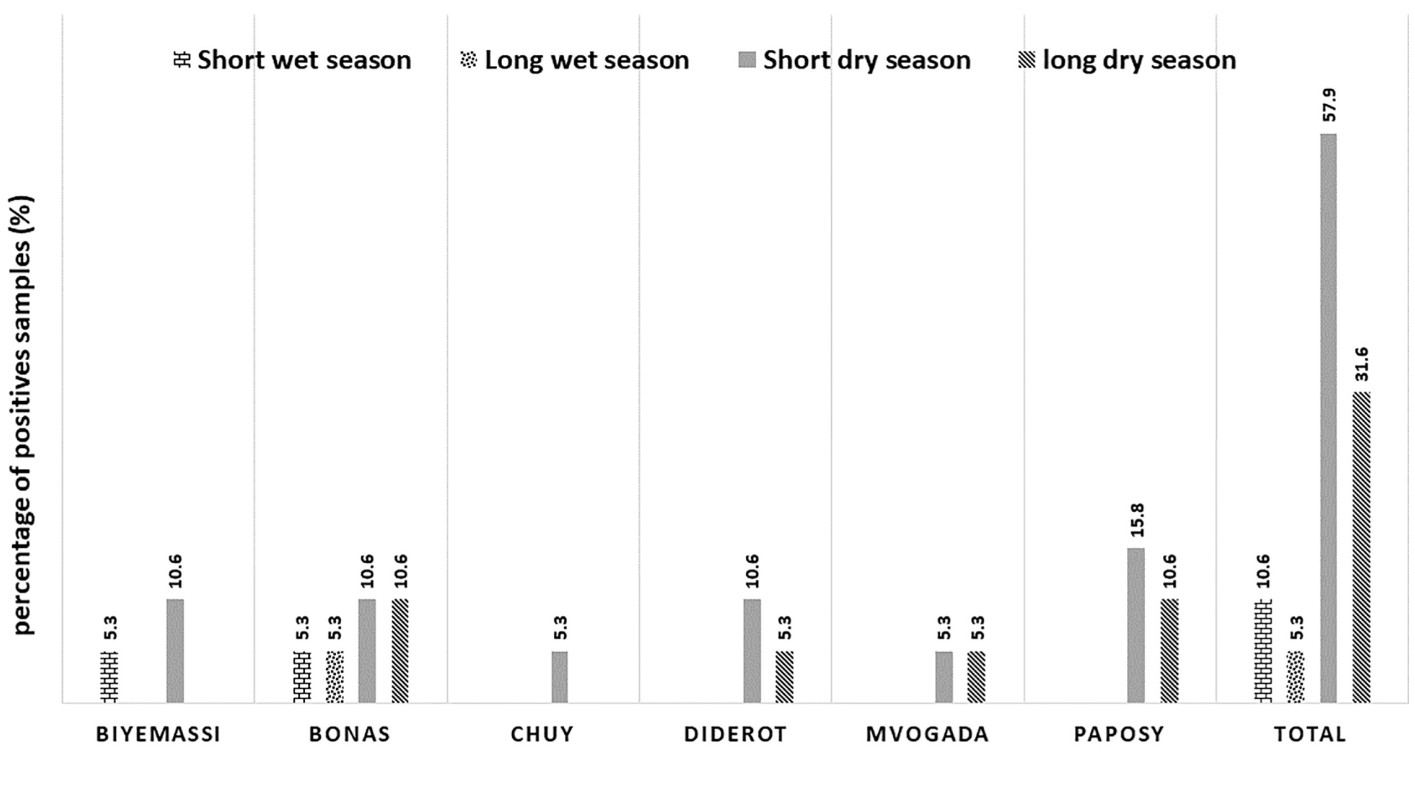

**Fig 1. Seasonal prevalence of HIV among sampling areas.**

This study detected varying degrees of HEV RNA in different geographic locations of sample sites. The present study found that wastewater from residential systems had the highest HEV detection rate, with a global detection rate above 70%. This may be due to the large number of people, the type of residence, poor hygiene conditions and a lack of a potable water supply, as well as population density. These findings suggest that overcrowding and poor sanitation increase the risk of HEV transmission through contaminated wastewater [1,28].

It is well known that surface waters from different sources are commonly used as crop irrigation sources worldwide, which could cause HEV contamination of irrigated crops [18,29]. In the present study, HEV RNA was detected at high levels in wastewater used for irrigation (Diderot) in the Mfoundi Division. The detection of HEV in irrigation water sources and plant samples should raise concerns for the public health system and local health authorities. Our result reveals a possible foodborne transmission through crops in this area and the wide distribution of HEV RNA in Diderot strengthens the importance of HEV environmental surveillance in Cameroon. Importantly, simple and low-cost measures can be implemented to reduce the risk of infection for communities in Cameroon.

Seasonal pattern of HEV positivity was described in the literature [25,30–34] but much less is known about seasonal influences of HEV distribution in Africa, and specifically in the countries with equatorial climate seasons. In this study, a seasonal pattern of HEV RNA positivity was observed in sewage samples collected over the course of a year in the Mfoundi Division, Yaoundé, Cameroon. A high detection rate of HEV RNA was observed at all collection sites during the short (36.8%) and long (31.6%) dry seasons, while a lower positivity rate was found during the long-wet season. These findings are consistent with previous studies on HEV detection in environmental water, providing evidence that changes

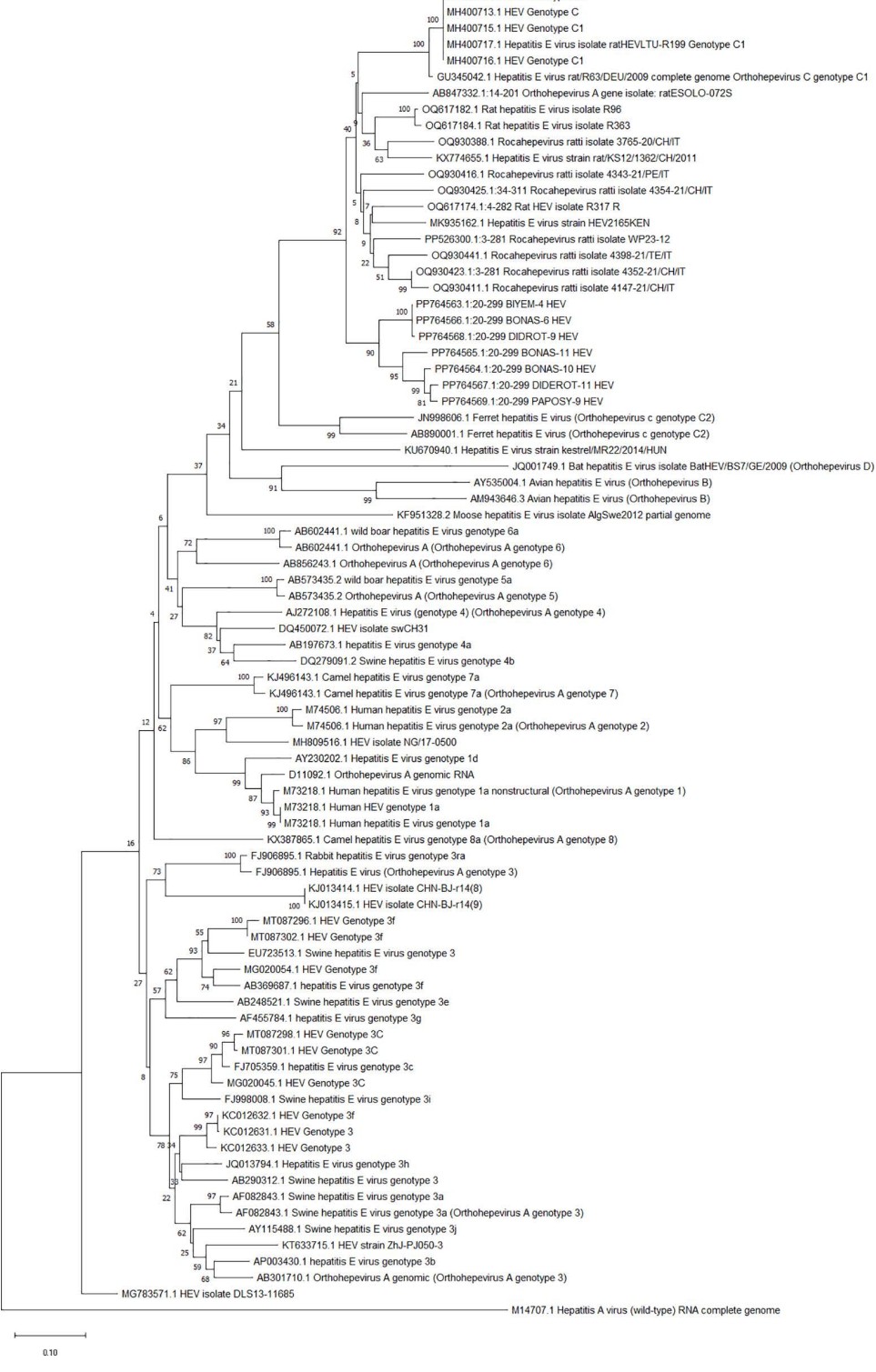

**Fig 2. Phylogenetic tree based on a fragment of the ORF1 region of GenBank sequences.** Only bootstrap values greater than 70% are shown. For each GenBank HEV strain included in the phylogenetic analysis, the accession number, host, genotype/subtype, and country of origin are provided. The strains identified in this study are outlined with a red circle and indicated by their accession number.

in climate conditions are linked to the rate of viral particle detection in environmental waters in Cameroon. [25,33]. This finding could be related to the fact that a reduction of water flow level may increase viral contamination when feces are discharged into the river. By contrast, our findings were markedly different from those reported in the Central African Republic where HEV was highly detected during rainfall season or flood in human samples from Bangui [34].

Our phylogenetic analysis of viral sequences from wastewater samples collected in Yaoundé revealed a clearly defined monophyletic cluster, suggesting that the viruses are genetically very similar. This points to the local circulation of a single variant or a group of closely related hepatitis E virus (HEV) strains. The clustering pattern aligns with what has been seen in other studies from low-sanitation environments, where wastewater reflects population-level virus circulation [35,36]. What stands out in our findings is that these sequences are more closely related to viruses from the Orthohepevirus C group particularly genotype C1 than to the classical human HEV genotypes (HEV-1 to HEV-4) [37]. Orthohepevirus C is typically associated with rodents and was, until recently, though not to infect humans. But that view has changed. A few recent studies from Europe have reported confirmed cases of human infection with HEV-C1, including in patients with acute hepatitis and weakened immune systems [38,39]. These cases suggest that this rodent-associated virus may be crossing into human populations more often than previously thought, and could be an emerging zoonotic threat.

Detecting HEV-C1-like sequences in wastewater from an urban setting like Yaoundé raises important public health questions. It's not yet clear whether these viruses are coming from infected people or from rodent populations shedding the virus into the environment. Either way, it highlights the need for stronger surveillance especially in crowded cities where contact with contaminated water or rodent reservoirs is more likely. Given these findings, there's a clear need to: carry out full-genome sequencing to confirm exactly which virus is circulating, explore possible animal reservoirs, particularly rodents in urban areas, and include environmental data like this in One Health strategies that link human, animal, and environmental health.

Our study has some limitations. The sample size was relatively small, and half of the sampling sites were residential areas, which might not fully represent the city's wastewater landscape. Still, even with these constraints, our data show that wastewater can be a powerful tool for tracking enteric viruses in the population and can offer early warning signals about potential outbreaks.

We also emphasize the need for further research, particularly involving comparative sequence analysis between HEV strains detected in wastewater and those from both human and animal origins especially local swine populations. Such investigations would be crucial to elucidate the role of zoonotic transmission and to deepen our understanding of the molecular epidemiology and transmission dynamics of HEV in the region.

## Conclusion

This study provides the first evidence of hepatitis E virus (HEV) detection and genetic characterization in wastewater from the Mfoundi Division in Yaoundé, Cameroon, and marks only the second report of environmental HEV in the country. HEV RNA was found in all six sampled sites, with particularly high detection rates in wastewater from residential areas and irrigation sources. These findings suggest that HEV is circulating widely in the community and that wastewater may be an important route of environmental exposure especially in areas with poor sanitation, overcrowding, and limited access to clean water. The presence of HEV RNA in irrigation water is particularly concerning, as it raises the possibility of food-borne transmission through contaminated crops. This reinforces the importance of environmental monitoring, even outside known outbreaks, as a way to detect early warning signs of viral spread in the population. Interestingly, our genetic analysis showed that the detected sequences are more closely related to Orthohepevirus C (genotype C1), which is typically found in rodents. This raises important public health questions about possible zoonotic transmission, particularly in urban settings where humans and rodents live in close proximity. Though traditionally considered non-infectious to humans, HEV-C1 has recently been linked to cases of hepatitis in Europe, suggesting it may be an emerging threat. While our study is limited by the number and type of sites sampled, it clearly demonstrates the value of wastewater surveillance in

tracking the spread of enteric viruses like HEV. Going forward, more extensive sampling and genome sequencing will be needed to confirm the viral sources and clarify whether transmission involves animal reservoirs especially local rodent and swine populations. These efforts will be key to understanding how HEV circulates in Cameroon and to designing effective public health responses grounded in a One Health approach.

The findings highlight new challenges in managing and protecting watersheds, given the potential for fecal contamination from both human and animal sources. Waterborne transmission of HEV is a public health concern that should lead to the inclusion of HEV in national public health surveillance protocols as a clinically relevant pathogen

## Supporting information

**S1 File. Inclusivity in global research.**
(PDF)

**S2 Table. Dataset.** Summarize Result of HEV from wastewater.
(PDF)

## Acknowledgments

We are grateful to the field staff from the Mfoundi Division who participated in identifying the study sites, and to the workers involved in collecting wastewater samples.

## Author contributions

**Conceptualization:** Dowbiss META-DJOMSI, Marie Atsama-Amougou.

**Data curation:** Dowbiss META-DJOMSI, Marie Atsama-Amougou.

**Formal analysis:** Dowbiss META-DJOMSI, Marie Atsama-Amougou, Modeste Romuald Ngamaleu.

**Funding acquisition:** Dowbiss META-DJOMSI, Marie Atsama-Amougou.

**Investigation:** Dowbiss META-DJOMSI, Marie Atsama-Amougou.

**Methodology:** Dowbiss META-DJOMSI, Marie Atsama-Amougou, Modeste Romuald Ngamaleu.

**Project administration:** Dowbiss META-DJOMSI, Marie Atsama-Amougou, Charles Kouanfack, Ahidjo Ayouba.

**Resources:** Dowbiss META-DJOMSI, Marie Atsama-Amougou.

**Software:** Dowbiss META-DJOMSI, Marie Atsama-Amougou, Modeste Romuald Ngamaleu.

**Supervision:** Dowbiss META-DJOMSI, Marie Atsama-Amougou.

**Validation:** Dowbiss META-DJOMSI, Marie Atsama-Amougou.

**Visualization:** Dowbiss META-DJOMSI, Marie Atsama-Amougou.

**Writing – original draft:** Dowbiss META-DJOMSI, Marie Atsama-Amougou.

**Writing – review & editing:** Dowbiss META-DJOMSI, Marie Atsama-Amougou, Celestin Godwe, Martin Maïdadi-Foudi, Marcel Tongo, Joseph Fokam, Charles Kouanfack, Ahidjo Ayouba.

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
