## [Decision Letter · Decision Letter 0]

14 May 2025

PONE-D-25-16759Molecular Epidemiology of hepatitis E virus in wastewater in Yaoundé, Cameroon: Toward optimal prevention and control of emerging zoonotic pathogensPLOS ONE

Dear Dr. META-DJOMSI,

Thank you for submitting your manuscript to PLOS ONE. After careful consideration, we feel that it has merit but does not fully meet PLOS ONE’s publication criteria as it currently stands. Therefore, we invite you to submit a revised version of the manuscript that addresses the points raised during the review process.

We look forward to receiving your revised manuscript.

Kind regards,

Janin Nouhin, Ph.D.

Academic Editor

PLOS ONE

2. Please include a complete copy of PLOS’ questionnaire on inclusivity in global research in your revised manuscript. Our policy for research in this area aims to improve transparency in the reporting of research performed outside of researchers’ own country or community. The policy applies to researchers who have travelled to a different country to conduct research, research with Indigenous populations or their lands, and research on cultural artefacts. The questionnaire can also be requested at the journal’s discretion for any other submissions, even if these conditions are not met.  Please find more information on the policy and a link to download a blank copy of the questionnaire here: https://journals.plos.org/plosone/s/best-practices-in-research-reporting. Please upload a completed version of your questionnaire as Supporting Information when you resubmit your manuscript.”

4. We noticed you have some minor occurrence of overlapping text with the following previous publication(s), which needs to be addressed:

https://d.docksci.com/first-evidence-of-the-hepatitis-e-virus-in-environmental-waters-in-colombia_59f0f730d64ab2a3b992cac0.html

In your revision ensure you cite all your sources (including your own works), and quote or rephrase any duplicated text outside the methods section. Further consideration is dependent on these concerns being addressed.

5. Please include your tables as part of your main manuscript and remove the individual files. Please note that supplementary tables (should remain/ be uploaded) as separate "supporting information" files.

6. We note that your Data Availability Statement is currently as follows: [All relevant data are within the manuscript and its Supporting Information files.]

7. When completing the data availability statement of the submission form, you indicated that you will make your data available on acceptance. We strongly recommend all authors decide on a data sharing plan before acceptance, as the process can be lengthy and hold up publication timelines. Please note that, though access restrictions are acceptable now, your entire data will need to be made freely accessible if your manuscript is accepted for publication. This policy applies to all data except where public deposition would breach compliance with the protocol approved by your research ethics board. If you are unable to adhere to our open data policy, please kindly revise your statement to explain your reasoning and we will seek the editor's input on an exemption. Please be assured that, once you have provided your new statement, the assessment of your exemption will not hold up the peer review process.   

8. Please amend your manuscript to include your abstract after the title page.

9. We note that Figure 1 in your submission contain [map/satellite] images which may be copyrighted. All PLOS content is published under the Creative Commons Attribution License (CC BY 4.0), which means that the manuscript, images, and Supporting Information files will be freely available online, and any third party is permitted to access, download, copy, distribute, and use these materials in any way, even commercially, with proper attribution. For these reasons, we cannot publish previously copyrighted maps or satellite images created using proprietary data, such as Google software (Google Maps, Street View, and Earth). For more information, see our copyright guidelines: http://journals.plos.org/plosone/s/licenses-and-copyright.

10. Please include a copy of Table 1 which you refer to in your text on page 5.

Reviewers' comments:

Reviewer's Responses to Questions

**Comments to the Author**

1. Is the manuscript technically sound, and do the data support the conclusions?

Reviewer #1: Partly

Reviewer #2: Partly

2. Has the statistical analysis been performed appropriately and rigorously? 

Reviewer #1: No

Reviewer #2: N/A

3. Have the authors made all data underlying the findings in their manuscript fully available?

Reviewer #1: Yes

Reviewer #2: Yes

4. Is the manuscript presented in an intelligible fashion and written in standard English?

Reviewer #1: Yes

Reviewer #2: Yes

5. Review Comments to the Author

Reviewer #1: General comments: This study describes the detection of HEV-3 in untreated wastewater collected monthly in 2023 in Yaoundé, Cameroon. The detection rate was 26.4% (19/72) of all sites, with significant differences in distinct locations and seasons. The study is well-designed, and the topic is important; however, some results are not appropriately presented. I have the following comments that the authors may consider.

Major comments:

The current title seems too big. Consider removing “Toward optimal prevention and control of emerging zoonotic pathogens”.

A brief introduction of genomic structure of HEV is suggested, since ORF1 gene has been mentioned in the main text. Additionally, the authors should also mention the classification of HEV-3 subtypes.

Statistical analysis has not been described in Material and Methods.

Figure 1: The texts are difficult to read. Please increase the resolution.

Table 1: What is the meaning of “positif”?

Line 126: This reviewer does not understand “In this study, 52.6% (10/19) of the samples showed a strong signal for HEV-RNA.” How did the authors determine a strong signal for HEV-RNA? Please specify this in more detail.

Figure 2 is difficult to comprehend. Percentage (%) should be at least indicated in the Y axis.

What is the subtype of these HEV-3 strains circulating in this region? Furthermore, the authors should conduct sequence analysis of HEV ORF1 fragments identified in this study. For example, the sequence identifies between them.

Figure 3: In the phylogenetic tree, rat HEV-C1 strains are very divergent thus unnecessary to be shown; in contrast, representative HEV genotypes within Paslahepevirus balayani can be included in the phylogenetic analysis. In addition, please indicate the HEV-3 subtypes on the right of the tree for a better understanding (see PMID: 29514938).

Discussion: As introduced in line 64, HEV-1 has been detected in Cameroon previously. Also, to the knowledge of this reviewer, both HEV-1 and HEV-2 are circulating in the neighbor country Nigeria (PMID: 30352598). The authors may discuss why no HEV-1/2 has been detected in the wastewater in this study.

Discussion, line 202: The sentence is overstated. The findings in this study does not support the hypothesis of zoonotic transmission of HEV in Cameroon, unless identical HEV sequences could be identified in both humans and animals. Please revise it.

Notably, references 19, 22, and 23 are identical! Please ensure that the references are properly cited.

Minor comments:

Line 47: Humans are also hosts of HEV-3. Please refer to the relevant review (PMID: 32810801).

Line 79: This sentence is figure legend which should be removed in the main text.

Line 92: Define “ORF”.

Line 93: The reference 2 is irrelevant.

Line 106: Change “readability” to “accuracy”.

Lines 110 and 115: “hepatitis E virus” has already been defined as “HEV” earlier.

Lines 112-114: Please define the abbreviations.

Lines 124-125: Remove this sentence which is a Table title.

Line 137: Remove this sentence of figure legend.

Line 140: It should be “Figure 3”.

Lines 148-151: Remove the figure legend in the main text.

Unify “HEV-3” and “HEV genotype 3” throughout the manuscript.

Line 214: Delete “(2)”.

Reviewer #2: This study illustrates the interest of wastewater based epidemiology for HEV, an enteric virus that can be transmitted zoonotically. Despite the low number of water samples analysed this study indicates the predominance of HEV genotype 3 circulation in Yaounde, Cameroon, rather than HEV genotypes 1 or 2 that are strictly found in human. This topic is interesting but the manuscript needs corrections and can be improved.

Comments

1. Methods

The region targeted in ORF1 gene for PCR detection and sequencing must be indicated. What is the limit of detection of the assay ? Wastewater can contain PCR inhibitors ; how was the analytical process controlled ?

2. Results, line 116 – 123

There are discrepancies between the text and table 1 : eg BONAS detection rate at 10.5% (2/19) line 116 and 41.7% (5/12) in Table 1; no corresponding site for 15.8% (3/19) line 116; repetitions lines 121-123

3. Results line 126 :

Is the signal intensity sufficiently reliable ? Was the signal intensity reproducible in duplicate PCR experiments ? Fig 2a is not very useful.

4. The detection rate according to the different sites could be related to the population size. A potential relationship should be analysed.

5. Phylogenetic analysis

Reference HEV sequences (Smith et al J Gen Virol 2020) must be used for determining HEV-3 subtype. This is relevant because HEV subtype 3a was previously detected in Cameroon and a comparison with HEV 3 subtypes detected in animals could provide additional informations

6. Discussion :

HEV RNA was detected in Yaounde wastewaters. The source is probably from humans infected with HEV but an animal source is also possible (eg infected pigs)? This point should be documented and discussed.

7. Discussion, lines 165-167 : Beside foodborne transmission of HEV, waterborne transmission is a possibility. However, the detection of HEV in wastewater does not prove waterborne transmission because this reflects only excretion of the virus from infected individuals.

8. References

All the references must be checked. Duplicate and triplicate were detected (eg ref 16 and 30 ; ref 19, 22 and 23 with confusion between names and surnames)

9. Figures and Table :

- Fig 1 legend must give more detail

- Fig 3 is missing

- Table 1 : a column with the sampling area should be added.

10. Minor points in introduction

- Line 41 : add contaminated food

- Line 47 : five instead of four

- Line 53 : HEV4 mainly found in Asia

- Line 55-56 : persistent infectious lead to liver cirrhosis (and not the reverse)

- Line 63 : add reference 16

6. PLOS authors have the option to publish the peer review history of their article (what does this mean? ). If published, this will include your full peer review and any attached files.

**Do you want your identity to be public for this peer review?** For information about this choice, including consent withdrawal, please see our Privacy Policy .

Reviewer #1: No

Reviewer #2: No

---

## [Author Response · Author response to Decision Letter 1]

2 Jul 2025

Thank you for the reminder. We have updated the manuscript to fully comply with PLOS ONE’s style requirements.

2. Please include a complete copy of PLOS’ questionnaire on inclusivity in global research in your revised manuscript. Our policy for research in this area aims to improve transparency in the reporting of research performed outside of researchers’ own country or community. The policy applies to researchers who have travelled to a different country to conduct research, research with Indigenous populations or their lands, and research on cultural artefacts. The questionnaire can also be requested at the journal’s discretion for any other submissions, even if these conditions are not met. Please find more information on the policy and a link to download a blank copy of the questionnaire here: https://journals.plos.org/plosone/s/best-practices-in-research-reporting. Please upload a completed version of your questionnaire as Supporting Information when you resubmit your manuscript.”

In this study, the PLOS questionnaire on inclusivity in global research was not used, as the research did not involve travel to a different country, engagement with Indigenous populations or their lands, or work with cultural artefacts. All research activities were conducted within the researchers’ own country and community.

Thank you for your comments regarding the inclusivity of global research. We appreciate the opportunity to provide more information. The completed questionnaire provides detailed information on our research practices and considerations has been included as a supplementary file in our revised manuscript. We have also completed method section lines 105-107 as follows: “Further information on scientific considerations relating to inclusivity in global research can be found in the Supporting Information. As a national reference laboratory for emerging and re-emerging epidemic preparedness and response, our work focuses on supporting public health efforts. No permits were required to access the field sites.” We would like to clarify the following points: (1) Research context: Our laboratory's role is to support national health efforts, and this research contributes to that goal. (2). Field site access: No permits were required for accessing the field sites, as wastewater samples were collected from public gutters in areas accessible to the general public. (3). Sampling activities: The sampling did not involve restricted or protected sites, and samples were not transported across state lines.

To conform to the Guidance of PLOS, we have added these sentence lines 105 to 107 as follows: As a national reference laboratory for emerging and re-emerging epidemic preparedness and response, our work focuses on supporting public health efforts. No permits were required to access the field sites. In addition, identification of the sample sites has been performed in collaboration with the staff of the Mfoundi Division. We have completed the Acknowledgements section lines x to y as follows: Acknowledgments: We are grateful to the field staff from the Mfoundi Division who participated in identifying the study sites, and to the workers involved in collecting wastewater samples.

4. We noticed you have some minor occurrence of overlapping text with the following previous publication(s), which needs to be addressed: https://d.docksci.com/first-evidence-of-the-hepatitis-e-virus-in-environmental-waters-in-colombia_59f0f730d64ab2a3b992cac0.html. In your revision ensure you cite all your sources (including your own works), and quote or rephrase any duplicated text outside the methods section. Further consideration is dependent on these concerns being addressed.

Thank you for bringing the minor overlap in text to our attention. We apologise for the duplication and have revised the manuscript accordingly. Specifically, we have rewritten the affected sections to eliminate redundancy and ensure originality. We have also thoroughly reviewed the manuscript to ensure that all sources, including our own work, are properly cited.

5. Please include your tables as part of your main manuscript and remove the individual files. Please note that supplementary tables (should remain/ be uploaded) as separate "supporting information" files.

Thank you for the guidance. As suggested, the table has been incorporated into the main manuscript on page 5, and the separate file has been removed.

6. We note that your Data Availability Statement is currently as follows: [All relevant data are within the manuscript and its Supporting Information files.] Please confirm at this time whether or not your submission contains all raw data required to replicate the results of your study. Authors must share the “minimal data set” for their submission. PLOS defines the minimal data set to consist of the data required to replicate all study findings reported in the article, as well as related metadata and methods (https://journals.plos.org/plosone/s/data-availability#loc-minimal-data-set-definition). For example, authors should submit the following data:

Authors do not need to submit their entire data set if only a portion of the data was used in the reported study. If your submission does not contain these data, please either upload them as Supporting Information files or deposit them to a stable, public repository and provide us with the relevant URLs, DOIs, or accession numbers. For a list of recommended repositories, please see https://journals.plos.org/plosone/s/recommended-repositories. If there are ethical or legal restrictions on sharing a de-identified data set, please explain them in detail (e.g., data contain potentially sensitive information, data are owned by a third-party organization, etc.) and who has imposed them (e.g., an ethics committee). Please also provide contact information for a data access committee, ethics committee, or other institutional body to which data requests may be sent. If data are owned by a third party, please indicate how others may request data access.

Thank you for your query regarding the data availability statement. We will revise it during the review process as follows: All of the data generated or analysed during the study are included in the article. The datasets used and/or analysed during the present research project are available from the corresponding author upon reasonable request. The viral sequences are available in GenBank.

7. When completing the data availability statement of the submission form, you indicated that you will make your data available on acceptance. We strongly recommend all authors decide on a data sharing plan before acceptance, as the process can be lengthy and hold up publication timelines. Please note that, though access restrictions are acceptable now, your entire data will need to be made freely accessible if your manuscript is accepted for publication. This policy applies to all data except where public deposition would breach compliance with the protocol approved by your research ethics board. If you are unable to adhere to our open data policy, please kindly revise your statement to explain your reasoning and we will seek the editor's input on an exemption. Please be assured that, once you have provided your new statement, the assessment of your exemption will not hold up the peer review process.

We apologise for this restriction and will review the data availability statement accordingly. “Data will be made available upon request to the corresponding author”.

8. Please amend your manuscript to include your abstract after the title page.

We apologise, we have already included our abstract after the title page (Lines 28 to 54) page 2.

9. We note that Figure 1 in your submission contain [map/satellite] images which may be copyrighted. All PLOS content is published under the Creative Commons Attribution License (CC BY 4.0), which means that the manuscript, images, and Supporting Information files will be freely available online, and any third party is permitted to access, download, copy, distribute, and use these materials in any way, even commercially, with proper attribution. For these reasons, we cannot publish previously copyrighted maps or satellite images created using proprietary data, such as Google software (Google Maps, Street View, and Earth). For more information, see our copyright guidelines: http://journals.plos.org/plosone/s/licenses-and-copyright.

Thank you for your note regarding the copyright of Figure 1. We confirm that the map was created using QGIS version 3.16.0 with administrative boundary data obtained from the Humanitarian Data Exchange (HDX), which is an open-source platform (https://data.humdata.org/). As the data from HDX is openly available, we believe the map can be published under the CC BY 4.0 license.

10. Please include a copy of Table 1 which you refer to in your text on page 5.

Thank you for the guidance. As requested, table 1 has been incorporated into the main manuscript on page 7, and the separate file has been removed.

Major comments: Molecular Epidemiology of hepatitis E virus in wastewater in Yaoundé, Cameroon: Toward optimal prevention and control of emerging zoonotic pathogens

The current title seems too big. Consider removing “Toward optimal prevention and control of emerging zoonotic pathogens”.

We agree with Reviewer #1’s comment regarding the length of the original title and support their suggestion to remove the phrase 'Towards optimal prevention and control of emerging zoonotic pathogens'. As requested, we have revised the title to: “Molecular surveillance of hepatitis E virus in wastewater in Yaoundé, Cameroon”.

A brief introduction of genomic structure of HEV is suggested, since ORF1 gene has been mentioned in the main text. Additionally, the authors should also mention the classification of HEV-3 subtypes.

We agree with Reviewer #1’s comment regarding the brief introduction of the genomic structure of HEV, the ORF1 gene, and the classification of HEV-3 subtypes. As requested and suggested, we have modified lines 62-74 of the “Introduction” as well as updated some references as follows: “The hepatitis E virus (HEV) is a non-enveloped, single-stranded, positive-sense RNA virus belonging to the Hepeviridae family, and the Paslahepevirus genus. Its genome is approximately 7.2 kilobases (kb) in length and comprises three open reading frames (ORFs). ORF1 encodes non-structural proteins involved in viral replication, ORF2 encodes the viral capsid protein, and ORF3 encodes a small multifunctional protein associated with virus egress and host interaction [9–11]. Among the eight identified HEV genotypes (HEV-1 to HEV-8), four (HEV-1, HEV-2, HEV-3, and HEV-4) are known to infect humans. HEV genotype 3 (HEV-3) is zoonotic and widely distributed among both humans and animals, particularly in developed and transitional countries. The primary animal hosts of HEV-3 include pigs, rabbits, deer, and mongooses. HEV-3 is further subdivided into multiple subtypes (e.g., 3a to 3j) based on phylogenetic differences. Similarly, HEV genotype 4 (HEV-4) infects both humans and pigs. Genotypes 3, 4, and 7 exhibit zoonotic potential. Genotypes 5 and 6 have been identified in wild boars, while genotypes 7 and 8 have recently been isolated from camelids and dromedaries [9,11]”.

Statistical analysis has not been described in Material and Methods.

We apologise for the absence of this part. In line with the guidance and as requested, we have added the relevant information on page 5, lines 142–145, in the 'Materials and Methods' section, as follows: 'Statistical analysis: Statistical analyses were performed using IBM SPSS Statistics 25.0 software. Qualitative variables were presented as frequencies, and the chi-square test was used to compare proportions, with a significance level set at p values <0.05”.

Figure 1: The texts are difficult to read. Please increase the resolution. Thank you for this observation.

We apologise for this. To comply with the reviewer's request, we have increased the resolution of Figure 1, as shown below:

Table 1: What is the meaning of “positif”?

We apologise for this misunderstanding. Here, 'positive' refers to detection; we have replaced it in lines 142 and 145 as well as in the table as follows:

lines 152-156: As shown in Table 1, HEV was detected in 26.4% (19 out of 72) of wastewater samples. Notably, HEV was identified in samples from all six study locations. The highest detection rates were observed in Bonas and Paposy, at 41.7% (05 out of 12 samples each), while the lowest rate was recorded in CHUY at 8.4% (one out of 12 samples). Biyem Assi and Diderot showed detection rates of 25% (03 out of 12 samples), while Mvog Ada's rate was 16.7% (two out of 12 samples).

Table 1: Summary of hepatitis E virus detection at each collection site

Sites N tested N detected % of detection

Biyem Assi 12 3 25

Bonas 12 5 41.7

CHUY 12 1 8.4

Diderot 12 3 25

Mvogada 12 2 16.7

Paposy 12 5 41.7

Total 72 19 26.4

N=Total number of samples, %= percentage,

Line 126: This reviewer does not understand “In this study, 52.6% (10/19) of the samples showed a strong signal for HEV-RNA.” How did the authors determine a strong signal for HEV-RNA? Please specify this in more detail.

We apologise for this misunderstanding. To clarify, the following details have been added to lines 165–170 on page 6: In this study, 52.6% (10/19) of the HEV-detectable samples exhibited a robust signal under UV illumination after electrophoresis, suggesting a higher concentration of HEV RNA. The signal intensity was assessed visually on the gel and scored on a scale from 1 to 3, where 1 indicates the weakest signal and 3 indicates the strongest. Of the samples that tested strongly positive, 30% (3/10) were from BONAS, 10% (1/10) from CHUY, 20% (2/10) from DIDEROT, 30% (3/10) from PAPOSY, and 10% (1/10) from BIYEMASSI.

Figure 2 is difficult to comprehend. Percentage (%) should be at least indicated on the Y axis.

Thank you for the remarks, as suggested, the detail of different percentages was added in the figure 2 as follows:

What is the subtype of these HEV-3 strains circulating in this region? Furthermore, the authors should conduct sequence analysis of HEV ORF1 fragments identified in this study. For example, the sequence identifies between them.

Thank you for your insightful comments on our manuscript. We appreciate your suggestion regarding subtypes circulating in this region. To address this issue, we have conducted further analysis of the sequences obtained in the present study. Our analysis revealed a high degree of similarity between our sequences and those of rat hepatitis E Orthohepevirus C genotype C1, as shown in Figure 3 below; The update was added on discussion, conclusion of the manuscript section line 234-283; and in result and conclusion of abstract line 40-51.

Figure 3: In the phylogenetic tree, rat HEV-C1 strains are very divergent thus unnecessary to be shown; in contrast, representative HEV genotypes within Paslahepevirus balayani can be included in the phylogenetic analysis. In addition, please indicate the HEV-3 subtypes on the right of the tree

---

## [Editor Report · Decision Letter 1]

11 Jul 2025

Molecular surveillance of hepatitis e virus in wastewater in yaoundé, cameroon

PONE-D-25-16759R1

Dear Dr. META-DJOMSI,

We’re pleased to inform you that your manuscript has been judged scientifically suitable for publication and will be formally accepted for publication once it meets all outstanding technical requirements.

Kind regards,

Janin Nouhin, Ph.D.

Academic Editor

PLOS ONE
---

## [Editor Report · Acceptance letter]

PONE-D-25-16759R1

PLOS ONE

Dear Dr. META-DJOMSI,

I'm pleased to inform you that your manuscript has been deemed suitable for publication in PLOS ONE. Congratulations! Your manuscript is now being handed over to our production team.

Kind regards,

on behalf of

Dr. Janin Nouhin

Academic Editor

PLOS ONE